# Reconsidering Ecological Civilization from a Chinese Christian Perspective

**Bryan K. M. Mok**

Centre for the Study of Religious Ethics and Chinese Culture, The Chinese University of Hong Kong, Shatin, N.T., Hong Kong; kiemanbryanmok@cuhk.edu.hk

**Abstract:** As part of the global effort to alleviate the ecological crisis, ecological civilization has become a dominant movement in China due to the state policy. Within this movement, the Chinese culture is said to be highly ecological and is thus an important asset to environmentalism. This paper seeks to offer a critical evaluation of this view by inquiring into its cultural and religious dimension with reference to Confucian and Chinese Christian thought. It argues that the construction of ecological civilization in China cannot rely only on the official discourses but requires a deeper cultural and religious investigation that helps realize the ecological potentials of the Chinese culture. In particular, it contends that the Confucian concepts of *qi* 氣 and *li* 理 can open up a way for humanity to attain unity with the cosmos and live in a path in harmony with nature through spiritual cultivation. It also suggests that the Christology and soteriology of Chinese Christian thinkers can strengthen this path of personal and social transformation by addressing the tendency of human beings to conform to selfish desire rather than the well-being of others. Both the Confucian and Chinese Christian worldviews are indispensable to the construction of ecological civilization by offering substantial insights into the cultural and religious dimension of the movement.

**Keywords:** ecological civilization; environmentalism; Confucianism; Chinese Christianity; sustainable development

---

## 1. Introduction

Civilization and nature are closely related to each other. Given the size of the contemporary human population, our activities are now generally considered as a major factor in climate change (Intergovernmental Panel on Climate Change (IPCC) 2013, p. 15; U.S. Global Change Research Program (USGCRP) 2017, vol. 1, p. 35). The thriving of our civilization has, no doubt, brought a substantial deterioration of natural ecosystems and resulted in a series of environmental problems, such as damage of marine habitats, decrease in crop harvest, and intensification of natural hazards (Intergovernmental Panel on Climate Change (IPCC) 2014, p. 4–8).[1] It seems that civilization is standing in an inverse relationship to nature. The question that follows is whether our civilization can be reconciled with nature.

The Paris Agreement, which was signed by 195 nations in 2015, was an attempt to reconcile civilization with nature. However, this attempt has been undermined by some political factors, such as Trumpism—which denies the reality of climate change and rates economic growth higher than biophysical well-being. However, there are still feasible measures for nations and states to reduce greenhouse gas emissions (United Nations Environment Programme (UNEP) 2017, p. xiv). Therefore,

---

[1]  IPCC releases its Assessment Report in a six to eight year interval. The latest available Assessment Report (the fifth one) was released in 2013–14, and the sixth Assessment Report is scheduled to be released in in 2021–22.

the question here is not whether we *can* do it, but whether we *will* do it. For this reason, as environmental scholar James Gustave Speth indicates, a spiritual and cultural transformation is needed on top of scientific and technological solutions to rectify human exploitation of nature (Wamsler 2019, p. 360).

To attain such a transformation, one of the tasks is to reconsider our relationship with nature by renewing the way we see and imagine the natural world (Cox and Pezzullo 2016, p. 12). In China, scholars have attempted to carry out this task by proposing the concept of ecological civilization since the mid-1980s. This concept is employed as a substitution of the anthropocentric and triumphalist attitude towards nature in industrial civilization with one that underscores human fundamental dependence on nature. It stresses the innate value of nature as well as the importance of a harmonious and symbiotic relationship between human beings and nature. In the past fifteen years, the Communist Party of China (CPC) has formally taken this concept into its official agenda and declared that it is the major principle of environmental policymaking. While this top-down ecological leadership may seem promising, the evidence of environmental devastation brought by Chinese industrial and economic activities may suggest otherwise. This calls for an examination and reconsideration of the rationale of ecological civilization.

Against this background, this paper is a philosophical investigation of the possible contributions of religion to the development of ecological civilization from a Chinese Christian perspective. Although the Chinese government has downplayed religion in the development of ecological civilization, religion does play a crucial role, as it offers comprehensive and normative pictures of the totality of the universe by means of symbolic representation (Conradie 2010, p. 273). The central argument of this paper is that while ecological civilization is closely related to Chinese religious traditions, an interreligious approach is necessary for a more meticulous understanding of the underlying worldview of ecological civilization.

## 2. The Concept of Ecological Civilization and Its Development in China

The concept of ecological civilization (*shengtai wenming* 生態文明) was first introduced by Chinese agricultural economist Ye Qianji 葉謙吉 in 1984 to advocate the development of ecologically sustainable agriculture, and was first regarded as a replacement of industrial civilization by ecological activist Roy Morrison in his book *Ecological Democracy,* published in 1995 (Pan 2016, pp. 34–35; Marinelli 2018, p. 373). It was then highlighted as one of the key political guidelines of the CPC in its 17th National Congress held in 2007 (Huan 2010, p. 200). The term was formally written into the constitution of China in 2018 (Hansen et al. 2018, p. 195). As we can see, ecological civilization—though not purely Chinese—is closely linked to the Chinese cultural and political context. Thus, it is important to review its reception and promotion in China when we are looking for a way to reconcile civilization with nature. However, before that, we need to have some basic understandings of the concept of ecological civilization.

According to Morrison (1995, pp. 9–12), the core of ecological civilization lies in its emphasis on the human embeddedness within nature. This reminds us that the flourishing of humanity depends on the conservation of a sustainable equilibrium of the biophysical sphere and hence calls us to live in a way that limits and transforms industrial civilization—an unsustainable civilization that is based on hierarchy, progress, and technique. Recently, some scholars have further developed this concept by highlighting both the discontinuity and the continuity between ecological civilization and industrial civilization (Gare 2010; Magdoff 2011; Pan 2016, pp. 29–50). According to them, ecological civilization parts ways with industrial civilization mainly in four ways. First, while industrial civilization shows a strong sense of reductionist materialism that instrumentalizes nature for human purposes, ecological civilization stresses the importance of symbiosis between human life and natural ecosystems. Second, ecological civilization rectifies industrial civilization's error of unrestrained economic growth by advocating self-regulation and self-restraint on human consumption. Third, it suggests that we should perceive the human relationship with nature in terms of harmony, not domination. Finally, it envisages that the goal of economic activities should no longer be output and profit maximization. Rather, we should achieve sustainable development.

Nonetheless, scholars have also stressed that we should not altogether perceive ecological civilization as a radical break from industrial civilization, for the former is continuous with the latter. First, as Chinese ecological economist Pan Jiahua (Pan 2016, pp. 42–45) suggested, ecological civilization has inherited the scientific knowledge and technologies emerging in industrial civilization. Industrial civilization has also united the world into a global civilization, which is the cornerstone of ecological civilization. Thus, philosopher Arran Gare (2010, p. 12) proposes that ecological civilization should not completely break away from previous civilizations, but should instead incorporate their best heritages into it. In other words, ecological civilization is a creative transformation rather than a total negation of industrial civilization.

Having had a brief overview of the concept of ecological civilization, we can now move on to review its reception in China. Ecological civilization is closely related to the Chinese social and cultural context in several dimensions. First of all, Chinese philosophy has been regarded as having a long tradition of ecological awareness, which is best demonstrated in the ancient concept of the "unity of Heaven and humanity" (*tian ren he yi* 天人合一). According to Daoist classic *Daodejing* (道德經, also known as *Laozi*, 老子), human beings shall be in harmony with the Way (*Dao* 道) of Heaven (*tian* 天), for their well-being is dependent on the latter (Pan 2016, p. 35). Several Chinese scholars share the view that the concept of the "unity of Heaven and humanity" is one of the most important resources for proposing a Chinese perspective of ecological ethics (She 2002, pp. 18–54; Qiao 2011, pp. 191–200; Wu 2012, pp. 85–123; Han 2013). In their works, Heaven is seen as a symbol that represents the totality of the universe. Thus, they believe that this emphasis on the unity of humanity and the universe can serve as a foundation of Chinese ecological ethics that balances the dualism between humanity and nature in Western thought. In turn, ecological civilization in China seeks to revive this philosophical tradition to create a sustainable future (Hansen et al. 2018, p. 197).

Secondly, a hint of environmentalism can be seen in socialism. For instance, Marx and Engels considered that industrial capitalism is an important source of unsustainable exploitation of nature (Heurtebise 2017, p. 10). In *Capital*, Marx (Marx [1867] 1990, vol. 1, p. 637) argued that capitalist production "disturbs the metabolic interaction between man and the earth." Engels also suggested that "harmony with laws of nature" is a necessary condition for human freedom (Magdoff 2011, p. 20). As socialism is the official ideology of China, sustainability and harmony with nature are often supposed to be guiding principles for the state in political decision-making.

This leads to the third point. As mentioned before, ecological civilization has been written into the constitution of China. Since 1995, sustainable development has entered into the agenda of the CPC and the Chinese government (Liu et al. 2018, p. 742). Then, at the beginning of this century, the Chinese government started to include sustainable development and ecosystem improvement into their process of policymaking. This was shown by the four targets set in the 16th National Congress of the CPC in 2002 and the call of Hu Jintao 胡錦濤—the then president of China—for improvements in ecological construction and education in 2005 (Pan 2016, pp. 47–48). At the same time, Chinese officials of environmental protection—most prominently, Pan Yue 潘岳—attempted to construct discourses for propagandizing ecological civilization with reference to the Chinese philosophical and religious tradition (Heurtebise 2017, pp. 7–8; Hansen et al. 2018, pp. 197–98). As mentioned, ecological civilization became a key political guideline of the CPC in its 17th National Congress in 2007 and was formally written into the constitution in 2018.

It seems that the long philosophical tradition of harmony and unity, the socialist ideology, and the state policy-making process have altogether put China at the forefront of ecological civilization. Some scholars such as Gare (2010, p. 6) suggest that "[t]he world should follow the lead of China" in advocating ecological civilization. However, we need to examine the official ecological discourses more carefully before making such a conclusion. While those discourses associated ecological thinking with the Chinese tradition rather uncritically, it would be better to have a closer look into the relationship between the ancient Chinese philosophy and the contemporary concept of ecological civilization before

being optimistic. In addition, we need to evaluate the reception of ecological civilization in China with an eye on its outcomes.

## 3. CPC's Official Discourses for Ecological Civilization

As an official of the CPC and a representing figure of the promotion of ecological civilization, Pan Yue (Pan 2003, 2006a, 2006b, 2007, 2008) has published several articles concerning the theoretical basis of ecological civilization. In those articles, he strived to prove that traditional Chinese culture can provide the intellectual resources that Western philosophies lack for the construction of ecological civilization. For him, Western civilization is fundamentally anthropocentric, and industrialization and capitalism have further made it unable to lead the world in environmental protection. In contrast, ecological conservation is an intrinsic part of Chinese culture and history, and Chinese civilization is inherently in line with ecological civilization. In particular, the Confucian principle of "unity of Heaven and humanity" has shaped Chinese civilization into one that respects the intrinsic value of nature and models ethical and moral teachings on nature. In addition, the Daoist principle that "*Dao* (the Way) follows what is natural" (*Dao fa ziran* 道法自然) has blurred the distinction between humankind and nature and brought about a unity of intersubjectivity between them, whereas Chinese Buddhism has upheld loving-kindness and compassion and taught that all forms of life possess Buddha-nature (buddhadhātu 佛性) and are with equal status. Pan also attempted to further justify his arguments by illustrating that Western environmentalism has turned to the East and to China to seek ecological wisdom.

In some ways, Pan can be seen as a spokesperson or theoretic explorer of the political agenda of ecological civilization of the CPC. Hu Jintao (Hu 2007, sct. 4) was the first political leader in China to officially adopt the concept of ecological civilization, but his description of it in his report to the 17th National Congress was rather brief. "*Shengtai wenming*"—the Chinese term for ecological civilization—just appeared twice in the whole report—in fact, in the same paragraph. However, the report explicitly spelled out the call for the construction of ecological civilization. In the report, Hu mainly focused on energy saving and the use of renewable energy, the structure of industrial enterprises and emission of pollutants, and the mode of consumption. More importantly, he requested the idea of ecological civilization to be firmly established in society. In his report to the 18th National Congress, Hu further elaborated his thoughts regarding the construction of ecological civilization. There were two major developments. First, he put the construction of ecological civilization into the overall framework of national strategy and linked it with economic, political, cultural, and social developments. Its goal is to "build a beautiful China." Second, he institutionalized ecological civilization and urged the state and the whole country to move forward to a "socialist ecological civilization" (Hu 2012, sct. 8). This direction largely remained unchanged in the current president of China Xi Jinping's 習近平 (Xi 2017) report to the 19th National Congress.

However, it is noteworthy that Xi (2019) published an article specifically on ecological civilization in *Qiushi* 求是—the main theoretical journal of the CPC. While this article was mainly a piece of propaganda that proclaimed how successfully the CPC and the state have advanced the construction of ecological civilization in these years and its significance to China, two points are worthy of attention. First, in his introduction of the article, Xi cited several passages from *I Ching* 易經 (also known as *Yi Jing* or the *Book of Changes*), *Daodejing*, *Mencius* 孟子, *Xunzi* 荀子, and *Qimin Yaoshu* 齊民要術 to illustrate the ecological heritage of Chinese civilization. While claiming that these Chinese classics underscored the unity of Heaven, Earth, and humanity and of the natural environment and human civilization, Xi did not give further analysis and interpretation of how they could contribute to the present construction of ecological civilization. Second, Xi articulated six principles that are necessary for the construction of ecological civilization. They are (1) symbiosis between humanity and nature, (2) the economic importance of a green environment, (3) ecological well-being as social welfare, (4) the interdependence of all beings in ecosystems, (5) environmental protection through the strictest legal rules, and (6) the endeavor for the construction of global ecological civilization. These two points, to some extent, show

Xi's ambition to propagandize the construction of ecological civilization as part of the CPC's political agenda by making use of traditional Chinese philosophy. However, it is quite obvious that he lacks a theoretical link or bridge between the agenda of ecological civilization and the Chinese tradition.

Pan Yue (Pan 2018) wrote another article that echoed Xi's agenda slightly earlier. The style of writing unambiguously tells that it was part of the propaganda campaign for Xi's leadership in ecological civilization. Thus, it was not surprising that Pan largely shared Xi's pattern of thought in that article. Pan attempted to further enrich Xi's thought by adding citations from the *Analects* and the *Master Lü's Spring and Autumn Annals* (also known as *Lüshi Chunqiu* 呂氏春秋) to demonstrate the ecological awareness in ancient Chinese culture. Like Xi, Pan did not explain how those passages of Chinese classics enlighten the contemporary construction of ecological civilization.

The above discussion shows that Chinese official discourses on ecological civilization have not filled a gap between the ancient philosophical ideas, such as the unity of Heaven and humanity, on the one hand, and modern ecology on the other. Philosopher Jean–Yves Heurtebise (2017, p. 9) has pointed out that the Heaven in classic Chinese thought refers to a universal law rather than the biophysical environment in the sense of modern ecology. Thus, the identity of the Heaven with nature in a biophysical sense may be an anachronistic over-interpretation. In fact, critics of Pan Yue's propaganda have already problematized this kind of appropriation of Chinese classics, which is said to be selective and reductionist (Hansen et al. 2018, p. 198). Moreover, despite the CPC's political rhetoric, China has not performed so well in environmental governance in recent decades. A recent study conducted by a group of Chinese scientists showed that during 1980–2010, the balance between human development and the natural environment in China has been greatly interrupted, though a slight improvement has been shown in the environmental friendliness of its economy. In any case, the study revealed that the level of ecological civilization decreased slightly (Zhang et al. 2016). According to general observations, moreover, it is hard to be optimistic that the environmental situation has dramatically improved in this decade, for we can see that the smog and air pollution are still very serious, and the threat of desertification has grown even stronger in China. It may be too hasty to conclude that the construction of ecological civilization in China is a failure. However, it clearly shows that officializing ecological civilization is one thing, but to bring actual ecological changes is another. What is needed here is a rationale that goes deeper into the cultural dimension that shapes people's worldview and way of living.

It is obvious that mere political rhetoric can hardly be a solution to environmental problems or a rationale that substantiates the construction of ecological civilization. Even worse, when ecological civilization is used as a political tool to support national ideologies, it will become an obstacle to environmental protection because it veils the genuine problems and prevents the taking of effective measures (Heurtebise 2017, p. 10). Furthermore, the ecological crisis is a global issue that cannot be sufficiently dealt with through a single cultural tradition. Instead, the construction of ecological civilization requires an international approach in both philosophical and scientific studies. Thus, a cross-cultural approach is essential for acquiring a full picture of an international understanding of ecological civilization and sustainable development (Liu et al. 2018, p. 747).

Finally, I would add that the construction of ecological civilization requires religious engagement. The transformation from industrial to ecological civilization is something as large as a paradigm shift. It not only involves an entirely different understanding of our relationship with the biophysical universe, but also calls people to radically alter their ways of living. This change in worldview and ethos is religious in nature, for it calls upon conversion of core beliefs and values. Lynn White, Jr. (White 1967) may have misunderstood the Christian tradition and been mistaken about the historical roots of the ecological crisis. However, to me, he was right in indicating the significance of religion for ecology and environmentalism. Thus, in the following section, I will discuss how religion may contribute to the construction of ecological civilization.

## 4. Religious Significance for Ecological Civilization

The foregoing discussion shows that political rhetoric alone cannot bring about a transformation in our worldview and way of living. At the same time, the sketchy appropriation of classical resources offers little help either. Instead of finding some apparent links between environmentalist concepts and ancient philosophy, we need to go deeper into the traditional wisdom in order to bring its transformative power to life. In short, we must explore the substantial dimension of culture. According to Paul Tillich (1959, p. 42), this dimension is what we call "religion."[2]

Some studies published in this journal have explicitly addressed how religions can and have contributed to the development of ecological civilization in China. For example, Yang and Huang (2018) found that most religions in China were favorable to public environmental behaviors, though the result in private environmental behaviors was just the opposite. Grounded on this empirical finding, they asserted that religions in China could provide an ethos that is fundamental to a civilizational change and also inform political decision-making in the process of ecological civilization. However, the question of how religion can lead to such a paradigmatic change in concrete terms remains untouched in their study. Like Yang and Huang, Chris Coggins (2019) also gave a positive account on the potential contributions of religion to the process of ecological civilization based on his ethnographical study of religious practices in Tibetan and Han villages. He also attempted to address how religion can have ecological contributions by illustrating that the Tibetan animism and the Han belief in *fengshui* have provided an ontological underpinning for the practice of ecological civilization.

Two other studies have attempted to deal with the particular contributions of individual religious traditions to ecological civilization by probing into the Daoist tradition. One of them is conducted by Jennifer Lemche and James Miller (Lemche and Miller 2019), who investigated how the Chinese Daoist Association (CDA) has promoted Daoism as a green religion to match up with the official agenda of ecological civilization. According to Lemche and Miller (2019, pp. 3–4), the CDA outlined four Daoist ecological teachings based on their main religious principles: Doing nothing against the principle "*Dao follows what is natural,*" conserving the harmony of yin and yang, restraining human activities from running counter to ecological balance, and stressing the importance of biodiversity in the light of the Daoist classics *Taipingjing* 太平經. However, they warned against any over-optimistic attitude towards the role of religion in China by remarking that the involvement of religions in ecological civilization is just another measure for the state to exercise authority over religious groups (Lemche and Miller 2019, p. 9). Another study by Martin Schönfeld and Chen Xia (Schönfeld and Chen 2019)—which investigates how Daoism has engaged in the state-coordinated agenda of ecological civilization—clearly reveals this limitation. This study is, to a large extent, conducted under the framework of the national policy of ecological civilization and attempts to fit Daoist teachings into such a framework. However, it did offer some substantial insights on religious environmentalism based on the lack of distinction between nature and culture in Daoism. According to Schönfeld and Chen (2019, pp. 6–8), this ambiguity underscores the ecological implications of human activities and the cultural implications of ecological changes. It then supplies a spiritual narrative, which is necessary for planetary stewardship—the task of human beings as a servant leader in healing the wounded planet.

While the above studies have shown some constructive ways in which religious traditions can contribute to the development of ecological civilization, it requires more critical reflections on the political framework imposed by the state, which is likely to compel religious groups to follow the state ideology and thus impede their prophetic role in ecological advocacy. In her paper, Lily Zeng (2019) attempted to uncover this problem with her case study of the Dai ethnic communities in Xishuangbanna,

---

[2]　Some Christian theologians—particularly those who follow Karl Barth—may disagree with this definition of religion, as it may undermine the distinctiveness of the Christian faith by being too close to culture. As I am not going to enter into a theological debate over this issue here in this paper, it suffices to say that from my point of view, the distinctiveness of the Christian faith—or any faith—is exhibited not by radically separating itself from culture, but by offering a profound perspective of culture.

Yunnan. Similarly to Lemche and Miller, Zeng reminded us that ecological civilization is indeed a way for the state to control and direct ethnic and religious minorities in the name of environmental protection and sustainable development. She further argued that political agenda always takes priority over environmental practices whenever there is a conflict. For her, this can be seen as a sugar-coated version of state control, which perpetuates Maoist ideologies.

The critical remarks of Lemche, Miller, and Zeng urge us to reconsider ecological civilization more cautiously. This does not mean that we have to abandon the entire concept or deny all religious engagement in it; but, while inquiring into the positive and constructive role of religious traditions in the development of ecological civilization, we also need to consider their prophetic and critical role. In other words, religion should not be merely the vassal or adjunct of the state authority, but should retain its critical position even if it conflicts with the political agenda. Thus, to widen the horizon, it is important to look beyond China in the discussion of religious contributions to ecological civilization.

Apart from following the official agenda, religious traditions should also examine human civilization with their own resources and from their own perspectives. Gary Gardner (2006, p. 3), for example, identifies the core problem of contemporary civilization as the lack of "a strong set of ethical boundaries that could sustain progress over the long term and orient it toward prosperity for all." Grounded on this view, he enumerates some assets of religion that can be useful to the transformation of civilization: The richness in symbols and rituals, the potential to generate moral and social capital, the number of its followers, and the physical assets (Gardner 2006, pp. 43–53). In particular, the symbolic and ritual power of religion can convey meanings that can hardly be expressed through ordinary language and experience, and thus have the potential to bring about an evolution of worldviews (Gardner 2006, pp. 20–21, 43–44). Environmental ethicist Pablo Martínez de Anguita (2012, p. xiii) shares the view that religion is an essential vehicle for the search for meaning and "proposes 'environmental solidarity' as the paradigm to sustain sustainability." For him, the most important task of religion is to offer answers to cosmological and existential questions that are concerned with the ontological dimension of environmental decision-making (Martínez de Anguita 2012, pp. 89, 140–41). In brief, the transformative power of religion—which Gardner calls a stimulus of worldview evolution and Martínez de Anguita thinks of as a vehicle for a paradigm shift—is invaluable for making human civilization more ecological.

Furthermore, not only can religion contribute to sustainable development, but sustainable development can also have a religious or spiritual dimension. Lucas Johnston (2013, p. 11) contends that besides scientific definitions, sustainability should also be understood as the mediation of "a brokering process between different constituencies, their epistemologies, and their visions of the good life." In turn, religion is important to sustainability in three ways: The provision of nature-as-sacred visions, the issuance of ecological pronouncements, and the emergence of a generic and humanistic civil religion. Each of these is not just a potential contribution, but "has consistently appeared in international political venues", such as the UN and World Bank (Johnston 2013, p. 54). All of these show that the incorporation of religious visions and values into an ecological agenda "facilitates sustainable relationships between people with different value structures" and hence helps the mediation of the brokering process, which is essential to sustainable development (Johnston 2013, pp. 2–3).

The above discussion illustrates that an important role of religion in environmentalism is its provision of symbolic representations and visualized pictures of the totality of the world. These representations and pictures are capable of bringing forth a transformation from an unsustainable to a sustainable way of living. They can take the form of stories, symbols, rituals, or even conceptual models. The key is that they connect people with different ethical, cultural, social, and economic backgrounds and call them together to turn against the idolatry of industrial civilization that has misled people into taking modern consumerism as the best way of living (Northcott 2009, pp. 226–29).

As the foregoing discussion has shown that ecological civilization—which upholds sustainability, symbiosis, self-regulation, and harmony—has a strong Chinese background. Although the development of ecological civilization in China is heavily influenced by state ideology and political agenda, we can

still consider it as a useful means "to give a moral legitimacy to the enforcement of environmental regulations" (Heurtebise 2017, p. 11). However, we need to put it under the scrutiny of different religious traditions—particularly their symbolic resources in representing the totality of the universe as well as the human relationship with it. I pay particular attention to symbolic representation because it lies at the heart of religion. It reveals the deepest aspects of reality (Eliade 1961, p. 12). In fact, religion is fundamentally a system of symbols that structurally guides men and women to understand their existence and hence establishes powerful and abiding moods and motivations in them (Geertz 1973, p. 90). Thus, I will attempt to push forward the discussion of ecological civilization in a cross-cultural approach by first surveying the symbolic representation of the human relationship with the universe in the Chinese tradition and then bringing the input of Christian theology into the discussion.

## 5. *Qi* and *Li*: Confucian Thought on Human–Cosmic Unity

Unlike Abrahamic religions, traditional Chinese thought has no idea of an external God who initially creates all that is and continuously keeps them in order. Contemporary Confucian philosopher Tu Wei-ming (Tu 1984, pp. 113–18) states that the Chinese model of the world is founded on the belief in the continuity of being. For the Chinese, the world is not a *creatio ex nihilo,* but a result of impersonal cosmic forces. In other words, existence requires no creator, and harmony needs no ordainer. For this reason, *qi*氣—the vital force of the cosmos—is an essential concept in understanding how nature operates. *Qi* brings everything into unity, for every being and entity in the world—whether living or non-living, heavenly or mundane—consists of and is made of it. Thus, in the depth of reality, humanity and nature are one. In turn, *qi* makes the world a closed system of self-generating life process. This system of process is not only an unbroken chain of being, but also a dynamic and organic whole. In the light of neo-Confucian philosopher Zhang Zai張載 (1020–77), the unity of Heaven and humanity gains its ecological meaning—humanity is the offspring of the Father Heaven and the Mother Earth, and they together form a holistic and organic body (Tucker 1998, pp. 191–97; Tu 1984, pp. 121–22).

On the one hand, human nature has a naturalistic side. Both humanity and nature are made of *qi* and are bodily connected through it. In this sense, humanity is a modality of *qi* (Dongfang 2005, p. 9). However, on the other hand, human nature is not just another product of *qi,* but has its uniqueness. Tu (1984, pp. 125–26) places this uniqueness of human nature in consciousness. With reference to the other neo-Confucian philosopher Wang Fuzhi's 王夫之 (1619–92) interpretation of the first chapter of the *Doctrine of the Mean*, Tu contends that human beings must keep enlarging and deepening their compassion in order to embody the whole cosmos. Only in this way can humanity fully follow the heavenly principle (*tianli* 天理). Seen in this way, humanity neither stands in a subject–object dichotomy with nature nor entirely merges into it. Rather, humanity forms one body with the myriad things in the cosmos through participation from within. The human mind, which is a refined and subtle form of *qi*, enables us to resonate with Heaven and Earth and the myriad things in it. In a later article, Tu (2002, p. 2)—by referring to classical Confucian philosopher Xun Kuang 荀況 (also known as Xunzi 荀子, 310–235 BCE)—further states that, while all things, including human beings, consist of *qi*, only human beings have a sense of righteousness. In this light, we can say that righteousness is a particular development of *qi* that shall be regarded as the overall principle of human activities and behaviors.

Tu's representation of the world as a self-generating life system reflects the Confucian worldview, which predominates over classical Chinese philosophy. In this worldview, the coming into being of all that is and the emergence of life is the result of the virtue of Heaven and Earth. It is a process similar to the growth of an organism (Lai 2020, p. 481). As mentioned, Tu considers that the human mind is a special form of *qi* that has righteousness, and so the responsibility of human beings is to act righteously by following the heavenly principle. In fact, in neo-Confucianism, this principle—or *li* 理—is closely connected with *qi*. In some sense, *li* can be seen as the neo-Confucian term of the ancient concept of *Dao* 道 (Kalton 1998, p. 82). While *qi* can be understood as the vital force of changes that runs throughout all that is and brings new things into being, *li* can be understood as the patterns of changes or the principle of *qi* (Tucker 1998, pp. 191–92). Michael Kalton (1998, pp. 83–85) argues that

the neo-Confucian concept of *li* can provide a profound philosophical interpretation of the systems theory emerging in the late twentieth century. Since *li* carries normative content for life, it can offer a moral dimension to various systems, such as biosystems, ecosystems, and social systems. In traditional thought, *li*—"a heaven-bestowed norm which is pure and perfect goodness"—is the cosmological root of ideal humanity. This idea can plausibly renew our understanding of the evolutionary process. Systems theory, according to Kalton, has already falsified the popular equation of evolution with "survival of the fittest." In fact, this is no longer regarded as the dominant pattern of the evolutionary process of life. To fit in the ecosystem, for example, symbiosis is indispensable because every existence in the world is in a web of responsive relationships with others. Human life cannot go on without the support of other forms of life as well as that of the surrounding ecosystems. *Li* reveals that the well-being of each of the bodies in the world is dependent on the well-being of all other bodies. It is thus the normative principle of the entire body of the cosmos and each of its parts. Accordingly, the development of ecological civilization hinges on how far we follow *li* and, hence, are in line with *qi*.

A question here is how we can know the way of *li* and follow through it. According to contemporary philosopher Dongfang Shuo 東方朔 (pseudonym of Lin Hongxing 林宏星), neo-Confucian Cheng Hao 程顥 (1032–85) claimed that all things in Heaven and Earth, including human beings, hold fast to *li*, which is the principle of perpetual growth and change (Dongfang 2005, p. 33). In other words, *li* is inherent in humanity just as it is inherent in the myriad things in the world. Dongfang (2005, pp. 57–82) further contends that Wang Shouren 王守仁 (also known as Wang Yangming 王陽明, 1472–1529)—commonly regarded as the synthesizer of neo-Confucianism—asserts that *li* never falls outside human hearts. In fact, in Wang's thought, it can be said that *li* is just another form of conscience. Furthermore, human conscience is in nature the same as the conscience of inorganic entities, for the myriad things are actually of one body. Those who are benevolent can get rid of their selfish desires and see themselves and the myriad things as the same body. In sum, the virtue of benevolence (*ren* 仁) is the key to connecting the human conscience with the cosmic principle, for it "constitutes some ontological continuity or unity between Heaven and humanity" (Lai 2020, p. 481). In other words, the boundary between divinity and humanity is not as strict as in traditional Chinese philosophy. To achieve the unity with the myriad things, one can appeal to one's conscience, for conscience unites humanity and divinity and will guide humanity to overcome selfish desires by means of the virtue of benevolence.

We have seen that the major problem of Chinese official discourses on ecological civilization, as shown in Xi Jinping's and Pan Yue's articles, is the superficial association of traditional Chinese culture and modern ecology. The concepts of *qi* and *li* can offer a rationale that helps ancient Chinese philosophy make sense in the contemporary context. For example, Chinese culture is said to be ecological not because some ancient Chinese texts seemingly supported the unity and harmony of humanity and nature, but because it offers a world-picture that reveals the commonality between human beings and other cosmic beings. Every act of us upon nature is an act upon ourselves, for the same *qi* flows in natural entities as in humankind. Hence, whenever we do good to the well-being of nature, we do good to our own well-being; whenever we harm nature, we harm ourselves. This cosmological picture not only gives a philosophical basis for the emphasis of symbiosis in ecological civilization, but also offers an enlightening way to perceive our innate interdependence and shared destiny with nature. In principle, Xi's official discourse is on the right track in highlighting the symbiosis between humanity and nature and the interdependence of all entities in the ecosystem. However, instead of simply making propagandist claims and enforcing environmental policies by law, it is more important to change the people's hearts. The principle of *li* in Confucian philosophy may help achieve this by appealing to the benevolence and righteousness that lie within our human nature. In fact, it is rather strange for Xi to uphold traditional Chinese culture on the one hand, while appealing to legal measures without any reference to human conscience and morality on the other. The (neo-)Confucian thought on *qi* and *li* reveals that real ecological change should begin not in the external legal system, but the natural law within (but not confined in) human conscience.

From the above discussion, we can see that Confucian philosophy has displayed an inward, self-referential, and humanistic understanding of human–cosmos relationship. One of the merits is its emphasis on human potentiality and possibility of perfection. From a Chinese perspective, the unity or oneness with the world and the myriad things in it is inherent in human nature. This is not to say that human effort can by any means surpass or replace divine actions, for natural phenomena are seen as results of the heavenly mandate. However, whether these phenomena lead to blessings or disasters is not unrelated to human morality. Therefore, for the well-being of humans and of the myriad things in the world, the crucial role of humanity lies in their moral and spiritual cultivation. One of the tasks of human beings in the cosmos is to assist the divine work of sustaining and nourishing life by actualizing the potential of humanity (Lai 2020, pp. 481–83). By recovering his or her "true self" through appealing to conscience and righteousness, one can break away from one's small self, which is occupied by selfish desires, and achieve a bigger self, which is in line with *li*. This perspective offers a positive way of understanding human responsibility for sustainable development and the possibility of constructing ecological civilization.

However, as I have argued above, this symbolic representation of the cosmic role of humanity needs to be complemented by other religious resources in the development of ecological civilization. To further investigate how religion can take up its role and make its contributions to ecological civilization, we need to study how other religious traditions represent the totality of the cosmos and the role of human beings. More importantly, we also need to question whether human beings, in reality, are able to turn away from selfishness and follow *li* in their living without any external assistance. If human beings can achieve harmony and unity with nature through activating their conscience, why does environmental degradation keep worsening? What hampers us to practice the benevolence and righteousness and follow *qi* and *li*? How can we overcome this predicament? The Christian tradition, which upholds the necessity of salvation from without, may provide a heuristic account that may further enlighten the construction of ecological civilization.

## 6. On the Necessity of an External Savior: A Chinese Christian Perspective

It is commonly known that Christianity is a faith that centers around Jesus Christ—the God–man who is held to be redeeming the world and humanity through his life and death. Thus, one of the major differences between Christianity and Confucianism (and also other Chinese religions) is the belief in the necessity of an external savior and redeemer. In particular, Christians believe that humankind cannot break away from fallenness without God's grace, which is ultimately revealed in Jesus Christ. Following the above discussion, the question here is how this worldview can further enrich the construction of ecological civilization in a religious dimension. While the analysis and interpretation of Confucian concepts of *qi* and *li* can provide an essential link between traditional Chinese culture and the contemporary construction of ecological civilization, its answer to the key to achieving human–nature unity is not yet sufficient because it lacks a way to tackle human sinfulness and fallenness. This is not just a matter of theory, but also one of reality, as shown in the seriousness of environmental degradation in China and elsewhere. In this sense, the Christian point of view is not something extra to the discussion of ecological civilization. On the contrary, it has the potential to rectify the major weakness of Confucian ideas. In fact, Confucianism is a long-time dialogue partner of Chinese Christianity. Therefore, a survey of how Chinese Christians have integrated Confucian thought into their theology may give us some further insights.

Both Roman Catholicism and Protestant Christianity are among the five officially recognized institutional religions in China, alongside Buddhism, Daoism, and Islam.[3] Unfortunately, the ecological account of Chinese Christianity is relatively thin compared with Confucianism, Daoism, and Chinese

---

3   Confucianism is not on the list because it is often not formally regarded as a religion, though it definitely offers a religious worldview. Nonetheless, Confucianism is recognized as one of the six major religions in Hong Kong.

Buddhism despite the global movement of ecotheology in recent decades. In addition, Christianity is often depicted as an unecological religion in the discourses of other religions in China. Nonetheless, in their endeavors to Sinicize Christian theology, Chinese theologians have demonstrated the possibility of constructing a Chinese version of Christian ecotheology. Religious scholar and theologian Lai Pan-chiu (Lai 2013, pp. 74–78) has provided a concise summary of those efforts. While Lai's account focuses on the theological concept of God of life, I will elaborate on the ecological potential of Chinese Christology in order to address the question of how Christian theology can complement the limitations of the Confucian concepts of *qi* and *li*.

The Church of the East (also known as *Jingjiao* 景教) was the first Christian church that reached China, and the inscription on the Nestorian Stele composed by Syrian missionary Jingjing 景淨 (eighth century) was the first Christian documentation in China, which survives to the present. In the inscription, Jingjing (Jingjing [781] 2009, pp. 19–20, 23) depicted Christ as God's alter ego divided from the Trinity. This triune God was said to be the ultimate mystery that creates all that is and generates Yin and Yang through the Holy Spirit. Human beings were commissioned to guard and cultivate nature, but Satan had made them turn away from their original nature and thus become unable to carry out the task. As a result, Christ was sent to redeem the fallen humankind, and he accomplished his mission by overcoming the plot of Satan through his death and resurrection. While employing Chinese symbolic representation in his theology, Jingjing differed from Confucianism in two ways. First, although all beings—including human beings—came into existence through Yin and Yang, they were created by a creator who transcends and precedes the world. Second, the way for human beings to return to the original goodness cannot be accomplished merely by self-cultivation, but radically depends on an external savior. From its very beginning, Chinese Christianity has been skeptical about any self-referential path of human–nature reconciliation.

These two primary differences between Chinese Christianity and traditional Chinese thought on human cultivation can also be seen clearly when Roman Catholicism entered China in the late sixteenth century. In answering the question of why there needs to be a creator when *qi* and *li* can explain the self-generation of all that is, Yang Tingyun 楊廷筠 (1557–1627)—one of the Three Pillars of Chinese Catholicism—argued that *qi* and *li* did not have consciousness and soul, respectively. Hence, *qi* and *li* alone are insufficient to explain the formation of the world, for *qi* is just an arbitrary force without a creator, and *li* only lies within matters, but cannot produce matters. The order of the universe and the possibility of life illustrates that every entity and organism in nature comes from a creator (Yang [1621] 2009, pp. 62–63). Interestingly, Yang (68–69) did not reject the neo-Confucian idea of the unity of nature and humanity. For him, the myriad things and the human subject are of the same body just as the neo-Confucian philosophers believed, but this did not mean that human beings and other entities in nature are of the same substance.[4] They are different in nature and yet the same in principle (*li*), different in faction and yet the same in origin. In this way, Yang (71) was able to preserve both the distinctiveness of humanity and its unity with nature. Concerning the necessity of a savior, he contended that only the redemption brought by Jesus through the cross could cancel out human transgression of the divine mandate. In other words, according to Yang's integration of Christianity and Confucianism, human beings inevitably transgress the natural order of *qi* and *li* thanks to their sinful nature. Only salvation through Jesus Christ could rectify this problem and enable humankind to follow *qi* and *li* once again. While Tu Wei-ming—as we can see in the above discussion—dealt with human unity-in-distinction with nature in a similar way, he—as a Confucian philosopher—did not find an external divine intervention necessary.

The link between Christ's redemption and the human relationship with the natural world was largely overlooked in Protestant Christianity in China—which was brought by western missionaries

---

4    In classical Chinese, the word *ti* 體 can mean both "body" and "substance." Yang attempted to differentiate these two uses of the word in his work.

who mainly focused on individual salvation—until the indigenous theology of the Three-Self Patriotic Movement—which was influenced by liberal theology—came on the scene. In particular, the concept of the cosmic Christ proposed by bishop and theologian Ting Kuang-hsun 丁光訓 (also known as K. H. Ting or Ding Guangxun, 1915–2012) attempted to reconnect Christology and soteriology with the wider universe. Ting's (Ting [1991] 2009, pp. 205–7) concept of the cosmic Christ has two main principles: The extension of Christ's reign, providence, and love over the entire universe and love as the core nature of Christ. Based on these two principles, Ting contended that Christ has been participating in the divine creation which has no end, and the goal of salvation is to make God's love, peace, and justice the principle of the cosmos. In the meantime, Christ is directing the process of history to that goal. Hence, salvation and redemption are not bound by the Christian church, but spread all over the universe. In turn, the whole universe is under God's providence and Christ's reign. In the light of Ting's concept of the cosmic Christ, which is heavily influenced by process theology, we can further infer that Christ has revealed the divine source of the vital force and the cosmic principle of the universe—or *qi* and *li*—and salvation and redemption allow humankind to properly take part in this cosmic order.

It is important to note that none of the Chinese Christians discussed above directly address ecology in the modern sense. Rather, their Christology only demonstrated some ecological implications and potentials. Among the notable modern Chinese theologians, Wang Wei-fan 汪維藩 (1927–2015) was perhaps closest to the construction of a conscious ecotheology. Grounded on the concept of *shengsheng* 生生 or "production and reproduction (of life)" in *I Ching* and the theology of the inscription of the Nestorian Stele,[5] Wang (Wang [1996] 2011b, pp. 106–10) depicted the Christian God as the source of *shengsheng* that keeps producing and sustaining life. Accordingly, Christ is the savior and fulfiller of life, and those who were redeemed by Christ shall see *sheng* (life) as the great attribute (*daide* 大德) and the highest excellence (*zhishan* 至善). Regrettably, Wang (Wang [1996] 2011a) did not further develop this insight in another article concerning the Christian faith and the ecological crisis. Nonetheless, Wang's interpretation of the Christian faith in terms of the perpetuating production and reproduction of life has provided important resources for reconsidering ecological civilization from a Chinese Christian perspective. If *qi* is the vital force of the cosmos and *li* is its principle, God is the ultimate source of life. God gives, fulfills, and sustains life in the cosmos by means of *qi* and according to *li*. Yet, sin has disabled human beings from following *li*, and hence caused distortions to *qi*. The impairment and loss of life due to environmental degradation by human activities is perhaps the most vivid demonstration of the effect of sin in our age. Humankind as a whole has been failed in carrying out their task of cultivating life. From a Chinese Christian perspective, Christ's redemption is the only way for humankind to return to the source of life and reconcile with *qi* and *li* by taking life—not only human life, but all sorts of life—as the top priority. *Shengsheng* can be resumed and life can go on only in this way. In other words, acknowledgement of the source of life and redemption by the God–man who has fulfilled and sustained life are necessary for ecological civilization.

## 7. Conclusions

The concept of ecological civilization has correctly pointed out that a deep transformation of our civilization is necessary if we want to preserve it. There are already plenty of signs—indeed, too many—pointing to the destruction of life as a consequence of our industrial civilization, which employs a laissez-faire approach to economic growth and excludes the well-being of other forms of life in nature. Showing our embeddedness in nature and dependence on other species, ecological civilization reminds us that we must acknowledge our symbiotic relationship with other beings in nature and take actions accordingly to restrain our economic activities in order to have a future.

---

5    The Chinese word *sheng* 生 can mean "to produce, generate, conceive, or give birth to" as a verb and "life" as a noun, and duplication often has a sense of recurrence or perpetuation. Therefore, *sheng sheng* primarily refers to the perpetuated activity of life.

Life and civilization can continue to flourish only if we take measures to replace profit maximization with sustainable development.

In these two decades, China has put ecological civilization into its official agenda of political decision-making. It has also attempted to integrate it with traditional Chinese culture and Marxist ideology to demonstrate its strengths and opportunities in environmental protection and sustainable development. However, its actual effect upon the alleviation of ecological destruction is yet to be seen, and some scientific research has prevented us from being too optimistic. A danger of the movement of ecological civilization in China is its focus on political propaganda rather than on a comprehensive investigation into the depth of culture. This lack of profound cultural analysis can be seen in the discourses of the political leaders, who were content with some superficial associations between environmentalism and Chinese classical works. To avoid being empty slogans, the construction of ecological civilization requires a closer look into the ecological rationale of Chinese religious philosophy.

Enjoying an official status throughout most of Chinese history, Confucianism has served as a backbone of the Chinese belief system. It may thus offer some important intellectual and spiritual resources for examining the idea of ecological civilization. Through their interpretation of neo-Confucian understanding of *qi* and *li*, the vital force of the cosmos and its principle, contemporary Confucians have established a link between human self-cultivation and ecological practices. According to them, we can attain the unity of humanity and nature by appealing to our conscience. In terms of ecological civilization, every one of us can overcome the selfish desire intensified in industrial civilization and market capitalism through moral and spiritual cultivation. This guides us to tune our way of living in accordance with *li* and thus attain harmony with *qi*. In a nutshell, the concepts of *qi* and *li* explain the spiritual reality of our embeddedness in the cosmic system and reveal a way for us to live in symbiosis with other beings in nature.

As Confucian thought has been incorporated into the Chinese culture in general and into other religious traditions in China, such as Daoism and Chinese Buddhism in particular, such a worldview and spiritual guidance is generally intelligible in China. However, the realities of pollution and environmental degradation in China have made the Confucian optimism about human cultivation dubious. Under the circumstances, the Christian insistence on the necessity of a savior from God does make sense. In their endeavor to indigenize the Christian faith, Chinese Christian thinkers of different ages have found ways to articulate their beliefs in God the creator and Christ the savior in Chinese terms. Their legacies may provide resources to supplement the limitations of the traditional Chinese worldview, which is often self-referential. The construction of ecological civilization requires human beings to restrain their economic activities and to sacrifice individual and collective wealth. Moreover, to cater to the well-being of other species means that our living may probably become less convenient and less comfortable—at least in the short run. In a negative way, Christian teachings remind us that humankind as a whole will always fail to do so without external assistance from God. It guards against any illusion that human beings can save themselves and the world from ecocide on their own. In other words, we need to confess our tendency to transgress *li* and hamper *qi*.

However, what positive contributions can this Chinese Christian perspective offer? Does it imply that one must first become a Christian before properly participating in the construction of ecological civilization? If this is the case, then this paper makes little sense in the public arena. If not, how can the Christological and soteriological inputs of Chinese Christianity complement and enrich our quest for a change in worldview corresponding to ecological civilization? To answer the above questions, it is important to note that salvation and redemption are not separated from creation in the Christian worldview. The doctrine of creation suggests that everything in the cosmos is from God and thus has a divine imprint. In the words of Pope Francis (2015, para. 76), creation "has to do with God's loving plan in which every creature has its own value and significance." In turn, the salvation and redemption through Christ was a renewal of this plan, which—as shown in Ting Kuang-hsun's account of the cosmic Christ—concerned the whole of creation. Seen in this way, the Christian emphasis on the prevalence of sin and the necessity of salvation does not negate the possibility of human perfection and

the endeavor to work for God's plan. In contrast, it proclaims the good news that, thanks to the life and death of Jesus Christ, "[a]ll of us can cooperate as instruments of God for the care of creation, each according to his or her own culture, experience, involvements, and talent" (para. 14), even though we are sinful. In this light, the Christian tradition has opened up a paradoxical interpretation of the unity of humanity and the cosmos. While affirming the human responsibility to respect the value of other forms of life and take care of their well-being, it always denies the possibility for us to attain symbiosis with other species and harmony with nature by our own effort. Besides encouraging us to try our best to do what we can do, this paradoxical perspective also reveals the importance of being critical of our own ecological practices and the environmental policies of our government. The official discourses on ecological civilization especially lack this critical perspective. Nonetheless, to a certain extent, this Christian perspective also justifies the necessity of legal measures in the construction of ecological civilization.

In principle, it is good for the CPC to take ecological civilization into its agenda, for a top-down approach may be highly effective, and effectiveness is important given the pressing urgency of the ecological crisis. If the construction of ecological civilization leads to actual improvement in environmental quality and a culture that treasures symbiosis and sustainability, China will be able to provide a successful model for the world to tackle the ecological crisis. However, there is still a long way to go. It requires radical changes in its mode of production and consumption, and these changes need the backup of profound ecological culture and spirituality. From a Chinese Christian perspective, whether China can achieve this is both yes and no. It is "yes" because balance, harmony, and unity are long regarded as the most important values in Chinese culture. In particular, Confucian thought has revealed the way to attain these values by self-cultivation. However, it is simultaneously "no" because human beings are always bound by selfish desire. Not only the Christian faith or the Bible says so, but the sheer fact of ecological destruction in China and the world also validates such a pessimistic view. Without realizing the no (our tendency to turn against what is good for both nature and humanity), we can never actualize the yes (our potential to attain unity with the cosmos). On the contrary, without affirming the yes (our task to cultivate life), we will easily fall prey to the no (despair and hopelessness). While ecological civilization may be just another propaganda for the CPC, it has the potential to constructively contribute to rectifying the fatal consequences of industrial civilization. Different religious traditions have a role to nurture—both constructively and critically—the best part of ecological civilization with their own spiritual wisdom. An insight that Chinese Christianity can offer is that while we always fail to follow the principle of the cosmic force that gives and sustains life, the renewal brought by the ultimate source of life has empowered us to work for the well-being of all life.

**Funding:** This research received no external funding.

**Conflicts of Interest:** The author declares no conflict of interest.

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
