# Peer review of "Reconsidering Ecological Civilization from a Chinese Christian Perspective"

_religions, doi:10.3390/rel11050261_

Round 1
Reviewer 1 Report
When I read the title and abstract, I was interested in reading the article. I have been in gatherings where the phrase "ecological civilization": was tossed around, amid references to China's leadership -- while at the same time seeing evidence of the ecological devastation caused by Chinese industry at home and abroad. If China were serious about this concept, it has world-changing implications; if it is just another propaganda ploy, it were best dismissed and not promoted in any way.
Two things are needed -- evidence of change, in Chinese industrial and ecological practices; and a rationale that shows it goes deeper than a line in a Party document.
This article has the potential to provide some interesting exploration of the rationale, but not in its present form. There are too many vague references to the term in the introduction, unconnected to the documents and how it is used in scholarship; and there needs to be more elaboration (given its usage) on how these might related to the concepts of qi and li that are (appropriately) cited. There is huge potential here, but more work needs to be done to make explicit what is only offered in this draft in outline.
The section, however, on Sallie McFague's work should be removed. There is no good reason to associate her ideas of a Christian, Protestant "God's body" and the concept of ecological civilization. In fact, only if there are equivalent theologies emerging from the Chinese Christian churches should the association even be attempted. It is my opinion that Christianity in (mainland) China has become an indigenous religion, thanks to the efforts of the Chinese government, especially under Chairman Mao, to exclude western religious influence. The result, ironically, has been to create a strong, independent and theologically distinct Chinese Christian church. It is to their theology that the author should look, to the expressions of ecological understanding of Creation and the relationship to other religious and philsophical traditions in China.
Reviewer 2 Report
There are several points where some further consideration could be given.
The IPCC references could be further checked in order to see whether the currency of the argument might be further assisted by any more recent reports. The IPCC is publishing progressively more reports.
The Chinese, Confucian, science / philosophical writings are consistently the most recent. That is not entirely true of the references to the Christian theological tradition. The dependence upon McFague (within a range of Christian writers) needs to be explained and further justified. The explanation at lines 499-502 is a little lightweight. There are many writers and some major publications like Conradie and Koster's, Christian Theology and Climate Change. I think it is reasonable to stay with McFagueso long as something further is added, explaining that the Christian theological positions are very diverse. Even a reference to the Pope's encyclical, Laudato Si' might be useful.
In some ways, the article is on the other side of a potential step-change in theological thinking. There is an increasing amount of work being done on how theology must engage with the dawning of the Anthropocene. How does this affect the argument based as it is on a distinction between industrial and ecological civilisation? I can see how this article could be the first of a couple of essays, the second focusing on the Anthropocene which presents some more demanding challenges.
I think the last paragraph of the introduction (p2, lines 48-54) are not necessary.
Who are the 'they', line 82.
Some theologians, especially those of a neo-orthodox perspective, would be critical of Tillich's understanding of 'religion' because it plays down (potentially) the distinctiveness of a revealed faith - indeed maybe any faith.
The sentence, lines 252-254, might be divided into two: it is really dense, abstract, and can end up masking what the writer wishes to say.
Line 459 perhaps it should be immanental rather than immanent
